# Immune Monitoring during Therapy Reveals Activitory and Regulatory Immune Responses in High-Risk Neuroblastoma

**DOI:** 10.3390/cancers13092096

**Published:** 2021-04-26

**Authors:** Celina L. Szanto, Annelisa M. Cornel, Sara M. Tamminga, Eveline M. Delemarre, Coco C. H. de Koning, Denise A. M. H. van den Beemt, Ester Dunnebach, Michelle L. Tas, Miranda P. Dierselhuis, Lieve G. A. M. Tytgat, Max M. van Noesel, Kathelijne C. J. M. Kraal, Jaap-Jan Boelens, Alwin D. R. Huitema, Stefan Nierkens

**Affiliations:** 1Princess Máxima Center for Pediatric Oncology, Utrecht University, 3584 CS Utrecht, The Netherlands; C.L.Szanto-2@prinsesmaximacentrum.nl (C.L.S.); a.m.cornel@umcutrecht.nl (A.M.C.); C.C.H.deKoning@umcutrecht.nl (C.C.H.d.K.); D.A.M.vandenBeemt@umcutrecht.nl (D.A.M.H.v.d.B.); E.Dunnebach-2@umcutrecht.nl (E.D.); M.Tas@prinsesmaximacentrum.nl (M.L.T.); M.P.Dierselhuis@prinsesmaximacentrum.nl (M.P.D.); G.A.M.Tytgat@prinsesmaximacentrum.nl (L.G.A.M.T.); M.M.vanNoesel@prinsesmaximacentrum.nl (M.M.v.N.); K.C.J.Kraal@prinsesmaximacentrum.nl (K.C.J.M.K.); A.D.R.Huitema-2@umcutrecht.nl (A.D.R.H.); 2Center for Translational Immunology, University Medical Center Utrecht, Utrecht University, 3584 CX Utrecht, The Netherlands; s.m.tamminga@amsterdamumc.nl (S.M.T.); E.M.Delemarre@umcutrecht.nl (E.M.D.); 3Division Imaging and Cancer, University Medical Center Utrecht, Utrecht University, 3584 CX Utrecht, The Netherlands; 4Stem Cell Transplantation and Cellular Therapies Program, Department of Pediatrics, Memorial Sloan Kettering Cancer Center, New York, NY 10065, USA; boelensj@mskcc.org; 5Department of Pharmacy and Pharmacology, Netherlands Cancer Institute, 1066 CX Amsterdam, The Netherlands; 6Department of Clinical Pharmacy, University Medical Center Utrecht, Utrecht University, 3584 CX Utrecht, The Netherlands

**Keywords:** neuroblastoma, immune monitoring, anti-GD2, IL-2, GM-CSF, ASCT, immunotherapy, dinutuximab

## Abstract

**Simple Summary:**

Neuroblastoma is a type of childhood cancer accounting for approximately 15% of childhood cancer deaths. Despite intensive treatment, including immunotherapy, prognosis of high-risk neuroblastoma is poor. Increasing amounts of research show that the fighting capacity of the immune system is very important for the outcome of neuroblastoma patients. Therefore, we investigated the fighting capacity of immune cells in blood at diagnosis and during the different phases of therapy. In this study, we observed both processes that stimulate and processes that decrease fighting capacity of immune cells in neuroblastoma patients during therapy. Despite this, we show that overall fighting capacity of the immune system of neuroblastoma patients is impaired at diagnosis as well as during therapy. In addition, we observed a lot of variation between patients, which might explain differences in therapy efficacy between patients. This study provides insight for improvement of therapy timing as well as new therapy strategies enhancing immune cell fighting capacity.

**Abstract:**

Despite intensive treatment, including consolidation immunotherapy (IT), prognosis of high-risk neuroblastoma (HR-NBL) is poor. Immune status of patients over the course of treatment, and thus immunological features potentially explaining therapy efficacy, are largely unknown. In this study, the dynamics of immune cell subsets and their function were explored in 25 HR-NBL patients at diagnosis, during induction chemotherapy, before high-dose chemotherapy, and during IT. The dynamics of immune cells varied largely between patients. IL-2- and GM-CSF-containing IT cycles resulted in significant expansion of effector cells (NK-cells in IL-2 cycles, neutrophils and monocytes in GM-CSF cycles). Nonetheless, the cytotoxic phenotype of NK-cells was majorly disturbed at the start of IT, and both IL-2 and GM-CSF IT cycles induced preferential expansion of suppressive regulatory T-cells. Interestingly, proliferative capacity of purified patient T-cells was impaired at diagnosis as well as during therapy. This study indicates the presence of both immune-enhancing as well as regulatory responses in HR-NBL patients during (immuno)therapy. Especially the double-edged effects observed in IL-2-containing IT cycles are interesting, as this potentially explains the absence of clinical benefit of IL-2 addition to IT cycles. This suggests that there is a need to combine anti-GD2 with more specific immune-enhancing strategies to improve IT outcome in HR-NBL.

## 1. Introduction

Neuroblastoma (NBL) is the most common extracranial solid tumor in children, accounting for approximately 15% of all pediatric oncology deaths [1]. Patients are stratified as low, intermediate or high risk (HR), depending on various factors (e.g., age, tumor stage, and several genetic components, such as MYCN amplification) [2]. HR-NBL patients are treated with multimodal therapy consisting of chemotherapy, high-dose chemotherapy followed by autologous stem cell transplantation (ASCT), resection of the tumor, local radiation, and maintenance immunotherapy (IT) consisting of the anti-GD2 monoclonal antibody, often combined with the cytokines IL-2 and GM-CSF, and isotretinoin acid [3,4,5]. Despite intensive treatment, 5 year event-free survival (EFS) is <50% [6,7].

Dinutuximab, the monoclonal antibody used in NBL IT, targets GD2 on the surface of NBL cells and signals antibody-dependent cell-mediated cytotoxicity (ADCC) and complement-dependent cytotoxicity (CDC) [3]. The rationale to alternately add GM-CSF and IL-2 to the IT cycles was to increase expansion and functional activity of natural killer (NK) cells, lymphocytes, monocytes/macrophages, and neutrophils. This was mainly supported by in vitro data indicating superior cytotoxic effects when combining dinutuximab with these cytokines [8,9]. Even though IT increased 2 year EFS and overall survival (OS) [3], relapses are still observed in the majority of patients.

The dose, timing, and chosen immunotherapeutic compound combinations are currently highly empirical and do not take patients’ immune status into account. Fast immune reconstitution during chemotherapy and higher absolute lymphocyte and monocyte counts have been associated with improved overall outcome in multiple cancers [10,11,12]. Nassin et al. showed that most patients with HR-NBL do not have full immune reconstitution at the start of IT (based on total white blood cell count (WBC), hemoglobin, and platelet, absolute neutrophil, lymphocyte and monocyte counts) and that immune recovery may correlate with disease-related outcomes [13]. Relatively fast NK-cell recovery early after ASCT was an important rationale for timing of IT early after transplantation [14]. Nonetheless, more detailed evaluation of NK-cell subsets showed that most cells are immature, cytokine-releasing (CD56bright, CD16+/−) rather than cytotoxic (CD56dim, CD16+). This may suggest suboptimal timing of dinutuximab IT early after transplantation, as cytotoxic NK-cells are mainly responsible for anti-GD2-dependent ADCC [13]. Nonetheless, to date, the potential effect of the IT regimen on shifting to the mature NK-cell phenotype has not been addressed.

Another important observation came from a phase III clinical trial where no additive effect of IL-2 administration on outcome of high-risk NBL patients was observed [4]. It is hypothesized that this may be the result of masking of the positive effects of IL-2 (e.g., on NK-cell expansion and functionality) by preferential regulatory T-cell (Treg) expansion [4,13], an effect known when administering (low dose) IL-2 to patients with autoimmune diseases [15]. Nevertheless, studies addressing this observation during NBL IT are lacking.

It may be hypothesized that post-ASCT immune reconstitution occurs with disparate kinetics in different patients, which may affect treatment efficacy of immune-targeting therapy. Comprehensive understanding of the status of the immune system in these patients may be instrumental for further development of immunotherapeutic interventions after ASCT. However, no studies have monitored the immune status in NBL patients during chemotherapy and IT and included functional analysis. Therefore, we monitored the immune status in NBL patients during chemo- and immunotherapy. In addition, the effect of IL-2 and GM-CSF on leukocyte and lymphocyte subpopulations and their (effector) cell functions during IT were studied.

## 2. Materials and Methods

### 2.1. Patients and Treatment

HR-NBL patients diagnosed between January 2015 and January 2018 treated in the Princess Máxima Center for Pediatric Oncology (Utrecht, The Netherlands) or Uniklinik Köln (Cologne, Germany) were included in this study. Patients were treated following the same treatment protocol based on N5/N6 chemotherapy (Dutch NBL2009 trial [16] and NB2013-HR pilot GPOH/DCOG trial; N5 = cisplatin, etoposide, vindesine, N6 = vincristine, dacarbacin, ifosfamide, doxorubicin). Staging was performed according to the International NBL Staging System (INSS) [17]. MYCN and ALK amplification status was determined with FISH, SNP-array was used for the determination of CNVs in 1p and 17q. The study was approved by the Medical Ethical Committees (Academic Medical Center, Amsterdam, the Netherlands; NL50762.018.14 and the University of Cologne, German trial 2013-004481-34). Written informed consent was obtained from the parents or guardians before enrollment in accordance with the Declaration of Helsinki.

### 2.2. Sample Collection

Peripheral blood samples (EDTA) were transported to the laboratory at room temperature (RT), and a Trucount cell subset enumeration tube was analyzed using flow cytometry within 24 h after blood withdrawal. Plasma was isolated after centrifugation and stored at −80 °C until analysis. Peripheral blood mononuclear cells (PBMCs) were isolated using Ficoll density gradient centrifugation, frozen in fetal calf serum (Bodinco, Alkmaar, The Netherlands) containing 10% dimethyl sulphoxide (Sigma-Aldrich, St. Louis, MO, USA), and stored in liquid nitrogen in the UMCU biobank until use in experiments. Frozen control donor PBMCs, taken from healthy adult volunteers, served as control group.

In Utrecht, peripheral blood samples were taken at diagnosis (1 sample from 7 patients), after each N5/N6 cycle (1–3 samples from 18 patients), before the high-dose (HD) chemotherapy regimen (1 sample from 7 patients), at start of IT (1 sample from 7 patients) and after 3 and 6 cycles of IT (1–2 samples from 8 patients) as depicted in Appendix A. In Cologne, peripheral blood samples were taken at start of IT and every 2 weeks during IT cycle 1–5. Samples were shipped at RT to the laboratory in Utrecht and processed within 24 h as described above.

### 2.3. Treg and NK-Cell Phenotyping

PBMCs were thawed and stained with either Treg or NK-cell discriminating antibodies. The Treg panel was comprised of the following extracellular antibodies: CD3-AF700, CD4-eFluor780, CD8-PE-Cy7, CD25-PE, CD127-BV421, CD45RO-BV711 (Biolegend, Biolegends, Koblenz; Germany). For intracellular staining, cells were permeabilized after extracellular staining, using the eBioscience kit (Thermo Fisher Scientific, Darmstadt, Germany) and stained for FOXP3 expression. The NK-cell panel comprised of CD3-AF700, CD19-eFluor780, CD56-PE-Cy7, CD16-BV510, CD45RO-BV711, TCRVα24-PE, TCRVβ11-FITC (Biolegend). All samples were measured within 24 h after staining on a BD LSR Fortessa (BD Biosciences, Heidelberg, Germany). All flow cytometry data were analyzed with FlowJo software version 10.6.0 (Tree Star, Ashland, OR, USA). Output CSV documents were further analyzed using RStudio (version 1.2.1335).

### 2.4. Proliferation Assay

To assess proliferation of T-cells, PBMCs were thawed, labelled with Celltrace Violet (CTV) (ThermoFisher Scientific)) and cultured in a round-bottom 96-well plate for 3 days at 37 °C and 5% CO_2_. 25,000 PBMCs were cultured in duplicates in the presence of anti-CD3 (0.5 μg/mL, 16-0037-81; ThermoFisher Scientific), or without stimuli. On day 3, supernatants were collected (pooled from duplos) and stored (as described in Section 2.6). Proliferation of PBMCs was analyzed using flow cytometry.

### 2.5. Suppression Assay

Patient and healthy-donor (HD) CD4+CD25highCD127low Tregs were sorted using BD FACSAria^TM^. Tregs were added to CTV-labelled effector cells at an effector-to-target ratio (E:T) of 2:1 in a crossover manner: (1) Tregs patient + effector cells patient; (2) Tregs patient + effector cells HD; (3) Tregs HD + effector cells patient; (4) Tregs HD + effector cells HD. Then, 96-well plates were coated with anti-CD3 (16-0037-81; ThermoFisher) to provide a proliferation stimulus. At day 3, the proliferation of effector cells was analyzed with flow cytometry.

### 2.6. Protein Profiling

Supernatant from the proliferation assays was collected after 3 days of culture, and stored at −80 °C until cytokine measurement. Interferon-γ (IFN-γ), tumor necrosis factor α (TNF-α), soluble IL-2R, IL-2, IL-10, IL-13, and IL-17 were measured using multiplex immunoassays (Luminex Technology, Austin, TX, USA). The multiplex immunoassay was performed as described previously by the MultiPlex Core Facility (MPCF) of the UMCU [18]. Out-of-range (OOR</OOR>) and extrapolated values were systematically replaced using the following procedure. The LLOQ (lower limit of quantification) and ULOQ (upper limit of quantification) were retrieved for the measured analytes of the experiment. The LLOQ and ULOQ values were retrieved per analyte by the MPCF. The lowest measurement was compared with LLOQ for each marker, to retrieve the lowest values for all measured markers. The same was performed for the highest value. OOR< data were replaced by the lowest value divided by 2. OOR> data were replaced by highest value times 2. The same procedure was performed for extrapolated data. For some markers, there are no LLOQ and ULOQ obtained yet. In that case, the lowest and highest measurements within the experiment were used for the replacement of OOR and extrapolated data.

Plasma samples were analyzed using the Proseek Multiplex Immuno-oncology immunoassay panel (Olink Biosciences, Uppsala, Sweden). Proseek is a high-throughput multiplex immunoassay based on proximity extension assay (PEA) technology that enables the analysis of 92 immuno-oncology-related biomarkers simultaneously. In short, PEA technology makes use of antibody pairs linked with matching DNA-oligonucleotides per protein of interest. These oligonucleotides hybridize when brought into proximity after binding the protein and are extended by DNA polymerase, thereby forming PCR targets. These targets are quantified by real-time PCR. Obtained results are expressed in normalized protein expression (NPX) values, which are in a log2 scale.

### 2.7. Statistics

Statistical analysis of absolute cell numbers and Treg expansion during IT was performed using the Mann–Whitney U test, comparing differences between groups before and after administration of IL-2 and GM-CSF. Hierarchical clustering analyses, presented as heatmaps, were based on Ward’s method and pairwise correlation distance. Heatmaps were generated using the heatmap.2 function from the gplots package [19]. To identify significant differences between protein levels before and after IL-2 and GM-CSF IT cycles, the Wilcoxon signed rank test was performed with correction for multiple testing according to Benjamini and Hochberg [20] for IL-2 cycles and the Mann–Whitney test with correction for multiple testing [20] for GM-CSF cycles. RStudio Project Software (version 1.2.1335) [21] was used for statistical analyses. Adjusted *p*-values of < 0.05 were considered statistically significant.

## 3. Results

### 3.1. Patient Characteristics

Twenty-five patients were included in this study (Table 1) with a median age at diagnosis of 3.9 years (range 0.3–10.8). A slight majority (56%, *n* = 14) had at least a partial response after induction chemotherapy. These patients continued therapy following the HR treatment protocol. Nonresponders (44%, *n* = 11) received additional chemotherapy (2–4 N8 cycles (etoposide, topotecan, cyclophosphamide)), and 14% (*n* = 4) received ^131^I-metaiodobenzylguanidine (^131^I-MIBG) therapy. Twenty out of 25 patients received HD chemotherapy followed by ASCT, seventy percent (*n* = 14/20) of patients received HD busulfan and melphalan (Bu-Mel) and 30% (*n* = 7/20) received HD carboplatin, etoposide, and melphalan. Following ASCT, 80% (*n =* 6/20) received dinutuximab IT in combination with cytokines. The four patients who did not receive IT had progressive disease. The mean time from ASCT to start IT was 137 days (range 108–193 days). The median time of follow-up for surviving patients was 2.14 years (range 0.65–3.67). The median event-free survival (EFS) was 1.65 years (range 0.11–3.67).

### 3.2. Immune Profiles at Diagnosis, during Induction Chemotherapy, and before High-Dose Chemotherapy Show Broad Variation between Patients

In the period before ASCT, large variations were observed between patients and between treatment cycles within individual patients in absolute leukocyte, lymphocyte, monocyte, neutrophil, eosinophil and specific lymphocyte subsets (B-cells, NK-cells, and T-cells) (Figure 1). Absolute neutrophil counts fluctuated most, peaking after the first N5/N6 chemotherapy cycle. B-cells decreased after the first round of N5/N6 chemotherapy and remained low during chemotherapy. Absolute lymphocyte counts remained similar between patients, while NK-cells and T-cells showed a large variation between patients. No correlation was found between absolute lymphocyte counts and occurrence of an event or MYCN status.

### 3.3. Immune Profiles during Immunotherapy Show Effect of IL-2 and GM-CSF on Leukocyte and Lymphocyte Subsets

To determine whether the in vitro effects of IL-2 and GM-CSF are also observed in vivo, immune profiles were generated during IT. In concordance with the rationale, total lymphocyte counts increased significantly after IL-2-containing IT cycles (*p* = 0.01), due to an increase of NK-cells (*p* < 0.01) (Figure 2 and Appendix A). IL-2 had no effect on total CD3+ T-cells (*p* = 0.67), CD19+ B cells (*p* = 0.70), and monocytes (*p* = 0.57). Neutrophils decreased significantly after IL-2 administration (*p* = 0.01), while eosinophils showed a trend towards increased numbers in peripheral blood after IL-2 (*p* = 0.19).

GM-CSF-containing IT cycles increased total lymphocytes (*p* = 0.05) and monocytes (*p* = 0.03), and a trend towards increased neutrophils (*p* = 0.07). GM-CSF had no effect on total CD3+ T-cell (*p* = 0.28), NK-cells (*p* = 0.12), and CD19+ B cells (*p* = 0.19) (Figure 3 and Appendix A). In addition, administration of GM-CSF resulted in a notable increase of eosinophils (*p* < 0.001).

### 3.4. Plasma Protein Profiling Further Supports IL-2 and GM-CSF Mediated Immune Engagement during Immunotherapy

Olink protein analysis was subsequently performed in plasma samples of 6 patients to determine protein profiles along the IT course. Protein profiling showed distinct patterns between pre- and post-IL-2 and pre- and post-GM-CSF-containing IT cycles. Unsupervised clustering resulted in complete separation of protein profiles pre- and post-IL-2-containing IT cycles (Appendix A) and partial separation of protein profiles pre- and post-GM-CSF-containing IT cycles (Appendix A).

Even though the sample sizes are too small to observe statistically significant differences upon IL-2-containing IT, increases can be observed in many NK-cell activation-associated markers, including GZMA/B/H, KIR3DL1, and NCR1 (all *p* = 0.18), IFN-γ (*p* = 0.34), CASP-8 (*p* = 0.17), and KLRD1 (*p* = 0.32) (Figure 4). Upon GM-CSF-containing IT cycles, significant increases in several neutrophil-, monocyte-, and eosinophil-associated factors, including CCL23 (*p* = 0.046), CCL17 (*p* = 0.015), CXCL11 (*p* = 0.037), and MCP-4 (*p* = 0.015) are observed (Figure 5).

### 3.5. NK-Cell Phenotype Varies Widely between Patients and Is Suboptimal for Efficient Dinutuximab-Mediated Cytotoxicity

As mentioned, the timing of IT in the NBL treatment protocol is established based on the observation of relatively fast NK-cell recovery early after ASCT [14]. Fast NK-cell recovery was observed based on absolute cell numbers (Figure 1g). However, even though variation is large, the balance between absolute numbers of mature, cytotoxic NK-cells (CD56^dim^CD16^+^) known to be mainly responsible for anti-GD2-dependent ADCC [13] and immature, cytokine-releasing NK-cells (CD56^bright^CD16^−^) was majorly disturbed at diagnosis and during all phases of the treatment protocol [22] (Figure 6a).

As plasma levels of NK-cell activation-associated markers increased upon IL-2-containing IT cycles, the NK-cell phenotype of two patients was subsequently assessed along the IT course. In both patients, we observed a major shift towards the mature, cytotoxic CD56^dim^CD16^+^ phenotype after both IL-2-containing IT cycles (Figure 6b,c). The CD56^dim^CD16^+^/CD56^bright^CD16^−^ ratio of patient 1 remained lower than the ratio of 9–9.5 in healthy controls [23], whereas the ratio of patient 2 reached a normal (IL-2 cycle 1) or superior (IL-2 cycle 2) NK-cell ratio.

### 3.6. Preferential Treg Expansion and Impaired T-Cell Proliferation during Therapy

Even though no significant changes were observed in absolute CD3+ T-cell levels after IL-2- or GM-CSF-containing IT cycles, it is suggested that cytokine therapy can shift the phenotype of CD3+ T-cells. To explore this effect during IT, extensive phenotyping of the CD3+ T-cell fraction was performed. Administration of IL-2 in this study massively increased the frequency of circulating CD4+CD25^high^CD127^dim^ FOXP3+ Tregs (Figure 7a,b). In addition, GM-CSF also increased the frequency of Tregs, although to a lower extent than IL-2 (Figure 7b). These data were supported by an increased trend in plasma levels of IL-10 (GM-CSF: *p* = 0.144, IL-2: *p* = 0.339) (Figure 7c).

To subsequently determine whether patient Tregs are functional, a Treg crossover suppression assay was performed in which patient Tregs from different IT time points were co-cultured with healthy-donor PBMCs. Healthy-donor PBMC proliferation was decreased upon co-culture with patient Tregs, indicating their suppressive capacity, even though suppressive capacity seems to be decreased when compared with healthy-donor Tregs (Figure 7d,e). In 2 of the 7 measurements (patient 1 cycle 2 and patient 3 cycle 1), no T-cell suppression was noticed.

To assess functionality of the CD3+ T-cell fraction in terms of proliferative capacity during IT, PBMCs were stimulated for three days with anti-CD3. Interestingly, anti-CD3-mediated T-cell proliferation was impaired in the majority of patients at different IT time points (Figure 8a). This was supported by decreased levels of secreted cytokines in stimulated patient PBMCs as compared to healthy-donor PBMCs (Figure 8b). Possible interference of CD25+CD127low Tregs or low-density eosinophils on T-cell proliferation was ruled out by performing additional T-cell proliferation assays without these cell populations.

## 4. Discussion

Absolute lymphocyte counts, relative presence of subsets, and their phenotypical characteristics are rarely monitored in NBL patients and not used as prognostic criteria or treatment guidance, largely due to a lack of knowledge on clinical significance. In the present study, we show that immune profiles of HR-NBL patients are already disturbed (reduced levels of CD3+, CD56+, and CD19+ lymphocyte subsets) at diagnosis when compared to age-matched controls [24]. This is in line with Tamura et al. [25], who also reported that lower levels of immune cells at diagnosis may predict poor prognosis in patients with NBL. As HR-NBL often disseminates to the bone marrow, it is hypothesized that the decreased immune cell levels are most likely caused by tumor replacement and/or by tumor-related suppressive factors present in the bone marrow niche [25,26]. This is supported by studies observing lower leukocyte [26] or monocyte and lymphocyte [25] levels in patients with bone marrow metastases.

Moreover, we confirm data from Chung et al. [26] showing that the decrease in total leukocytes and lymphocytes in children with HR-NBL is even more pronounced after chemotherapy. We however observed a large interpatient variability between chemotherapy cycles; while B cells are completely depressed during all stages of N5/N6 chemotherapy, the numbers of monocytes, NK and T lymphocytes differed enormously. Whether these variations correlate to clinical outcome will be subject of follow-up studies with larger cohorts.

The effect of chemotherapeutic agents on the immune compartment should be kept in mind when combining IT with re-induction chemotherapy in relapsed/refractory patients. The effect of chemotherapy on IT efficacy is paradoxal, as levels of effector cells are often affected. On the other hand, targeting of immunosuppressive immune subsets and increased immunogenicity of tumor cells are described as processes to enhance IT efficacy [27,28,29]. Timing and chemotherapeutic compound selection are key to maximize the effect of IT in refractory/relapsed patients.

When subsequently looking into the functionality of T-cells at diagnosis and during the therapy regimen, we noticed hampered proliferation and cytokine secretion upon anti-CD3-mediated T-cell stimulation. In line with this, impaired PHA mitogenesis at diagnosis and during NBL therapy has been observed in several studies [30,31]. Helson et al. [31] and Pelizzo et al. [32] showed hampered PHA-mediated T-cell mitogenesis when cultures were supplemented with serum of NBL-patients, or mesenchymal stromal cells (MSCs) from HR-NBL patients, respectively. This indicates the presence of both local and systemic immune modulation by the NBL tumor. Several factors have been described that are able to modulate T-cell functionality, including TGF-β, Indoleamine-pyrrole 2,3-dioxygenase (IDO), and arginase [33,34]. The depletion of arginine by arginase [33] leads to T-cell cycle arrest, impaired proliferation, and reduced activation [35,36]. Although impaired T-cell proliferation is already noticed at diagnosis, it should be noted that immune function may be further inhibited by intensive treatment. In-depth phenotyping, proteomics, and pathway-analysis of T-cells during HR-NBL treatment is necessary to unravel mechanisms responsible for T-cell dysfunctionality as a first step to develop strategies to counteract this effect.

The effect of the IT regimen on NK-cell phenotype is largely unknown. Even though variation between patients is considerable, our data indicate that the cytotoxic NK-cell ratio increased during IT. We observed a delayed increase of the cytotoxic ratio in two patients upon IL-2-containing IT cycles. However, the NK-cell phenotype ratio of the majority of patients is still decreased at the end of IT, which suggests suboptimal IT timing. The observed differential effect of GM-CSF- and IL-2-containing IT cycles on the cytotoxic NK-cell ratio indicates that this is an effect induced by IL-2 rather than dinutuximab itself.

To our knowledge, this is the first study to show beneficial effects of GM-CSF and IL-2 addition to IT cycles in HR-NBL patients on both NK-cells (increased cytotoxic NK-cell ratio and plasma levels of NK-cell-associated factors (e.g., granzymes, KLRD1, NCR1, IFN-γ, CASP-8, KLRD1)), as well as on myeloid cells (based on plasma levels of neutrophil/monocyte-associated factors (e.g., CXCL11, CCL17, CCL23, and MCP4)). Nonetheless, Ladenstein and colleagues [4] recently concluded from a phase III clinical trial that there is no additive effect of IL-2 administration on outcome of HR-NBL patients. We noticed a strong increase of CD127dimCD25highFOXP3+ Tregs after IL-2, and to a lesser extent, also GM-CSF administration. This increase has been described before [37]; however, in many cases without confirming FOXP3 positivity, this may be expected based on results from autoimmune patients [15] where (low dose) IL-2 is administered to induce Tregs. Previously, preclinical data showed that Tregs inhibit anti-NBL immune responses before and after ASCT [38,39,40]. Using functional suppression assays in a crossover format, we showed that these Tregs also maintain their suppressive capacity at multiple time points during IT. Together, these data suggest that the beneficial effects of IL-2 may be masked by preferential Treg expansion.

The observation of increased NK-cell cytotoxicity during IL-2-containing IT cycles in our opinion substantiates the need to replace IL-2 during dinutuximab IT with other non-Treg engaging (immuno)therapeutic compounds/strategies to maximize IT efficacy. First of all, the start of IT can be delayed to allow further recovery of the NK-cell fraction. However, the observation that the NK-cell phenotype is already disturbed at diagnosis, together with the risk of the tumor to expand before the start of IT, are arguments against postponement of IT. A second strategy would be to combine dinutuximab with soluble factors more specifically activating NK-cells, for example, Lirilumab, an anti-KIR antibody currently tested in the ESMART trial from the ITCC (ClinicalTrials.gov Identifier: NCT02813135). In addition, NKTR-214, a CD122-biased cytokine agonist designed to preferentially activate and expand effector CD8+ T- and NK-cells over Tregs via the heterodimeric IL-2 receptor pathway (IL-2R-βɣ) [41], is an interesting candidate to replace IL-2 [42]. Combining dinutuximab with IL-15 is also of interest, as this cytokine is known to specifically expand and mature NK-cells, without affecting Treg expansion [43,44]. The delayed effect of IL-2 on the cytotoxic NK-cell ratio observed in this study may substantiate an approach in which NK-cell engaging therapy is provided prior to dinituximab-based IT. A third strategy would be to combine IT with an adoptive NK-cell therapy at the start of IT to maximize effector cell function, either via an autologous (ClinicalTrials.gov Identifiers: NCT02573896, NCT04211675) or allogeneic (haploidentical) [45] strategy (ClinicalTrials.gov Identifier: NCT03242603). The advantage of using allogeneic cells is the potential to select a mismatched donor to maximize anti-tumor effect. On the other hand, the risk of graft rejection and mismatch-related adverse events in allogeneic settings is a clear disadvantage compared to the use of an autologous, ex vivo-expanded, cell product.

Immune monitoring of HR-NBL patients comes with some limitations. The availability of patient samples was limited by dropout of patients from the study after relapse/progression of disease, transfer to other trials, failure of blood sampling, and logistical issues. In this study, immune status was monitored in peripheral blood only, which provides markers that would be easily translatable to monitoring protocols in the clinic. Nevertheless, information on tumor-infiltrating lymphocytes (TILs), and monitoring lymphocytes in tissues, would help to elucidate the mechanisms of (resistance to) therapy, and indicate whether markers at the tumor site are systemically reflected in the blood. Multinational collaborations in NBL cohorts are needed to allow for a larger sample size to confirm the findings from this study and relate them to clinical parameters and outcome.

## 5. Conclusions

(Functional) immune monitoring in HR-NBL patients revealed the presence of both immune-enhancing and immune regulatory effects during the therapy course. The immune-enhancing effects observed upon IL-2-containing IT cycles, despite simultaneous Treg expansion, clearly demonstrate the potential of combining dinutuximab with other NK-cell engaging strategies. In addition, the observed systemic T-cell dysfunction at diagnosis as well as during HR-NBL therapy highlights another mechanism, besides lack of MHC-I expression and immune checkpoint expression, that should be unraveled to generate long-term anti-NBL immune responses and immunological memory needed to prevent relapse.

## Figures and Tables

**Figure 1 cancers-13-02096-f001:**
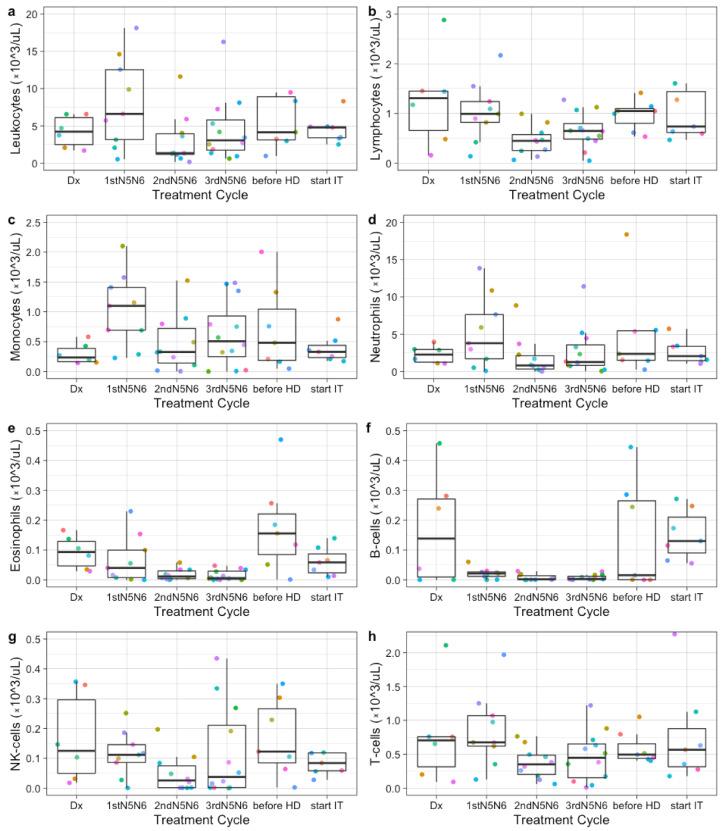
Immune profiles at diagnosis, during induction chemotherapy, and before high-dose conditioning. Each colored dot indicates absolute counts from one patient (×10^3^/uL). Absolute leukocyte (**a**), lymphocyte (**b**), monocyte (**c**), neutrophil (**d**), eosinophil (**e**), B cell (**f**), NK cell (**g**), and T cell (**h**) numbers are shown at diagnosis (Dx), after the 1st, 2nd, and 3rd round of N5/N6 induction chemotherapy, before high-dose chemotherapy (before HD), and at start of immunotherapy (start IT) from 6, 9, 10, 12, 7, and 4 patients respectively.

**Figure 2 cancers-13-02096-f002:**
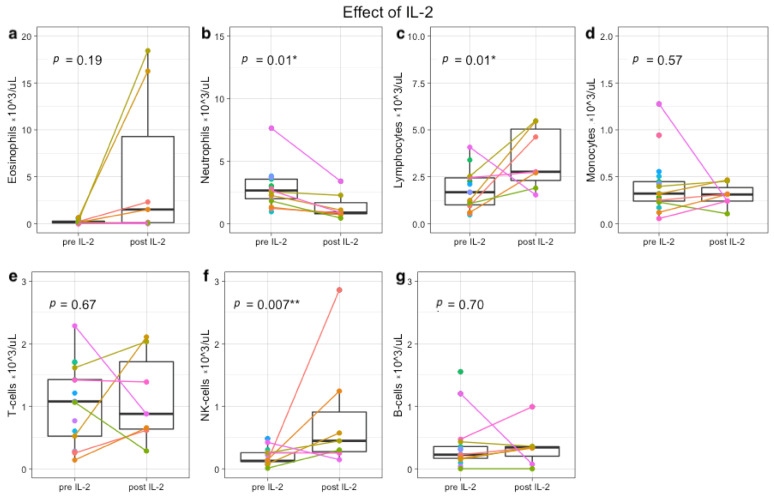
Immune profiles before and after IL-2-containing immunotherapy cycles. Each colored dot indicates absolute counts from one patient (×10^3^ cells/uL). From 5 patients, samples were paired before IL-2 (day 1 IT cycle 2 or 4) and after IL-2 (day 15 IT cycle 2 or 4). In total, 7 paired samples are depicted (colored lines), because two patients were monitored in both IL-2 cycles. Nine single measurements from 9 other patients were included, resulting in a total of 14 patients (11 in study, 3 leftover material during IT). Absolute eosinophil (**a**), neutrophil (**b**), lymphocyte (**c**), monocyte (**d**), T-cell (**e**), NK-cell (**f**), and B-cell numbers (**g**) are shown. * *p* < 0.05, ** *p* < 0.001.

**Figure 3 cancers-13-02096-f003:**
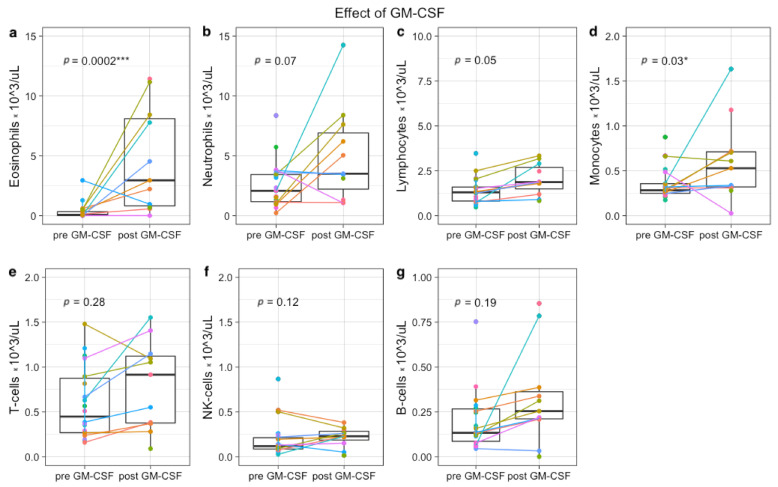
Immune profiles before and after GM-CSF-containing immunotherapy cycle. Each colored dot indicates absolute counts from one patient (×10^3^ cells/uL). From 5 patients, samples were paired before GM-CSF (day 1 IT cycle 1, 3 or 5) and after GM-CSF (day 15 IT cycle 1, 3 or 5). In total, 9 paired samples are depicted (colored lines), because two patients were monitored during all 3 GM-CSF cycles. Twelve single measurements from 12 other patients were included, resulting in a total of 17 patients (11 in study, 6 left over material during IT). Absolute eosinophil (**a**), neutrophil (**b**), lymphocyte (**c**), monocyte (**d**), T-cell (**e**), NK-cell (**f**), and B-cell numbers (**g**) are shown. * *p* < 0.05, *** *p* < 0.0001.

**Figure 4 cancers-13-02096-f004:**
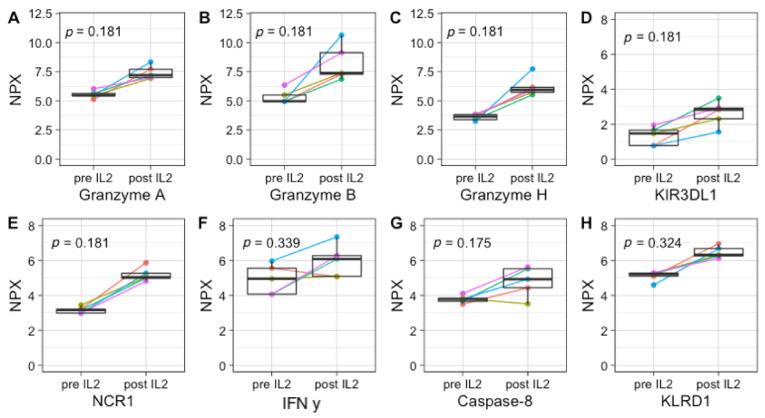
Upregulation of NK-cell activation-associated protein markers upon IL-2-containing immunotherapy cycles. Plasma protein concentration of GZMA/B/H (*p* = 0.181) (**A**–**C**), and KLRD1 (*p* = 0.324) (**D**), NCR1 (*p* = 0.181) (**E**), IFN-y (*p* = 0.339) (**F**), CASP-8 (*p* = 0.175) (**G**), and KLRD1 (*p* = 0.324) (**H**) pre- and post-IL-2-containing IT cycles. Protein expression is shown as normalized protein expression (NPX). In total, 5 paired samples are shown, as two patients were monitored during both IT cycles.

**Figure 5 cancers-13-02096-f005:**
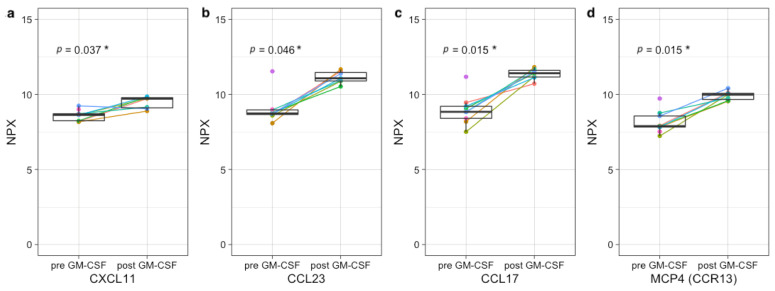
Upregulation of neutrophil-, monocyte-, and eosinophil-associated factors upon GM-CSF-containing immunotherapy cycles. Plasma protein concentration of CXCL11 (*p* = 0.037) (**a**), CCL23 (*p* = 0.046) (**b**), CCL17 (*p* = 0.015) (**c**), and MCP-4 (**d**) (*p* = 0.015) pre- and post-GM-CSF-containing IT cycles. Protein expression is shown as normalized protein expression (NPX). In total, 7 paired samples are shown, as two patients were monitored during all three IT cycles. Two single measurements from patients pre-GM-CSF were included, resulting in a total of 9 patients pre- and 7 post-GM-CSF. * *p* < 0.05.

**Figure 6 cancers-13-02096-f006:**
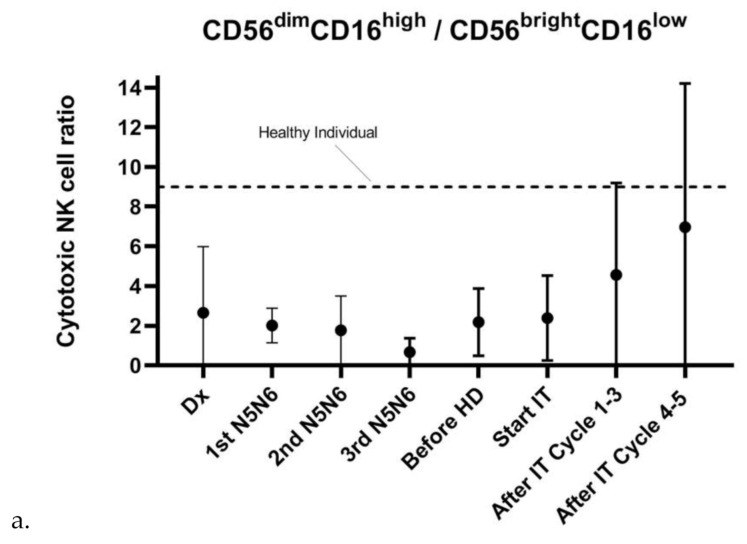
The cytotoxic CD56^dim^CD16^+^/CD56^bright^CD16^−^ NK-cell ratio during HR-NBL therapy. (**a**) The ratio of absolute CD56^dim^CD16^+^ and CD56^bright^CD16^−^ Trucount cell numbers is highly variable between patients and is decreased at diagnosis and during therapy of HR-NBL patients. Dx: *n* = 7, 1st N5/N6: *n* = 11, 2nd N5/N6: *n* = 10; 3rd N5/N6: *n* = 11, before HD: *n* = 7, start IT: *n* = 7, After IT Cycle 1–3: *n* = 10, After IT cycle 4–5: *n* = 8. The dotted line reflects the reference value of the cytotoxic NK-cell ratio of healthy individuals [22]. (**b**,**c**) In-depth monitoring of the fraction of CD56^dim^CD16^+^ and CD56^bright^CD16^−^ in two patients during the IT course shows an increase in cytotoxic (CD56^dim^CD16^+^) NK-cell phenotype after IL-2-containing IT cycles. In patient 1, the ratio remains below the normal cytotoxic NK-cell ratio of 9, whereas the ratio of patient 1 reaches normal values after the first IL-2-containing IT cycle and is increased after the second IL-2-containing IT cycle. Red arrows indicate start of IL-2-containing therapy cycles.

**Figure 7 cancers-13-02096-f007:**
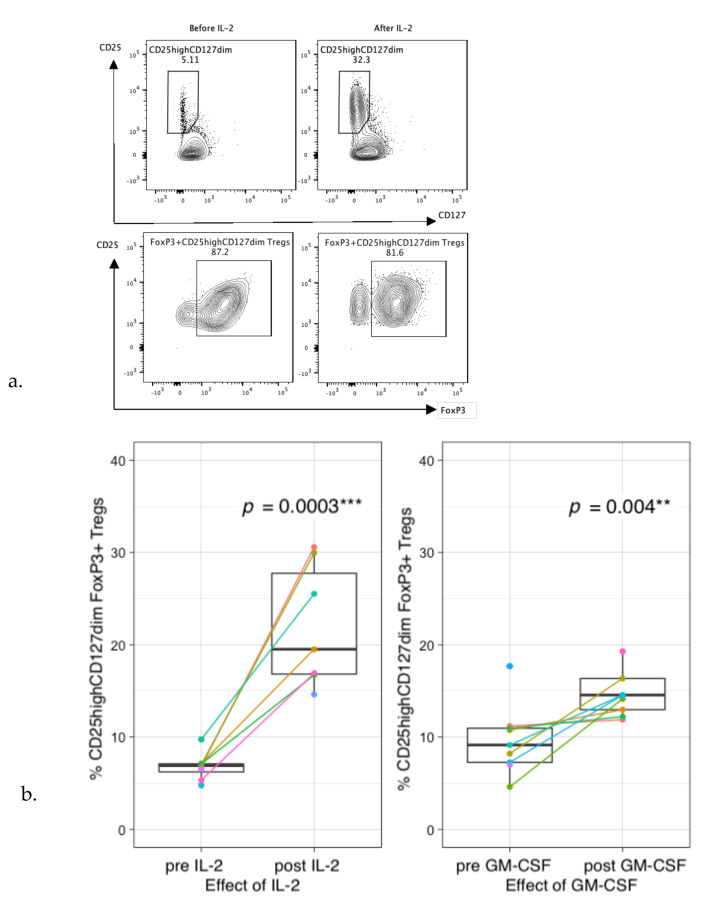
Regulatory T-cell profiles and their suppressive capacity during immunotherapy. (**a**) Example of gating of CD25^high^CD127^dim^ cells within the CD3+CD4+ T-cell population (upper panels) and gating of FoxP3 within the CD25^high^CD127^dim^ cell population before and after IL-2 administration (lower panels). (**b**) Percentages of Tregs (within CD3+CD4+ T-cell population) increase 4–5-fold after IL-2 administration (left) and increase 1–2-fold after GM-CSF administration (right). (**c**) Plasma IL-10 levels pre- and post-IL-2 (*p* = 0.339) (left) and GM-CSF (*p* = 0.144) (right). Protein expression is shown as normalized protein expression (NPX). IL-2: In total, 5 paired samples are shown, as two patients were monitored during both IT cycles. GM-CSF: In total, 7 paired samples are shown, as two patients were monitored during all three IT cycles. Two single measurements from patients pre-GM-CSF were included, resulting in a total of 9 patients pre- and 7 post-GM-CSF. (**d**) CTV staining of PBMCs of a healthy donor co-cultured without Tregs (grey), with patient Tregs (green), or healthy-donor Tregs (blue), or unstimulated (red) at an effector-to-target ratio of 2:1. (**e**) Relative percentages of proliferation of HD CD3+ T-cells co-cultured with patient Tregs (blue) or HD Tregs (green) compared to proliferation without Tregs (red). CD3+ T-cell proliferation was measured in patient 1 (during cycle 2 and 4), patient 2 (during cycles 1, 2 and 5) and patient 3 (during cycle 1 and 2). HD = healthy donor, PT = patient. ** *p* < 0.001, *** *p* < 0.0001.

**Figure 8 cancers-13-02096-f008:**
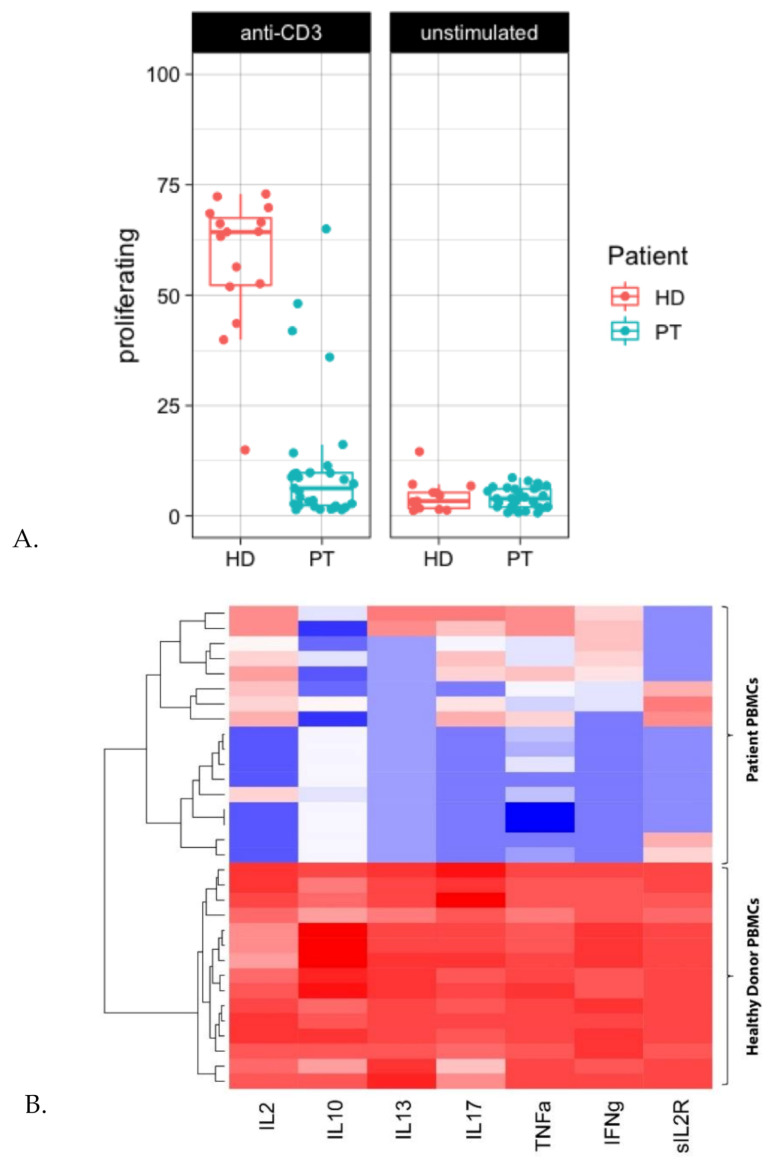
T-cell proliferation is impaired at diagnosis as well as during therapy in HR-NBL. (**A**) PBMCs of healthy donors (HD) (red) and patients (PT) (blue) were stimulated with anti-CD3 (0.5 µg/mL). T-cell proliferation of each individual sample is shown (duplos were pooled); PBMCs HD (*n* = 8), PBMCs patients (*n* = 12). (**B**) Supernatants (HD *n* = 15, patients *n* = 17) were analyzed using Luminex-based multiplex immunoassays. The heatmap shows the log concentration of IL-2, IL-10, IL-13, IL-7, TNF-α, IFN-γ and soluble IL-2R, with low levels indicated in blue and high levels indicated in red.

**Table 1 cancers-13-02096-t001:** Patient characteristics and time of sampling.

Patient Characteristics	Total (*n* = 25)
**Gender** male female	14 (56%) 11 (44%)
**Median age at diagnosis, year**, (range)	3.9 (0.3–10.8)
Stage 3 disease	1 (4%)
Stage 4 disease	24 (96%)
***Genetics***	
**MYCN** Neg Gain Amp	18 (72%) 2 (8%) 5 (20%)
**1p** normal partial loss loss gain	14 (56%) 9 (36%) 1 (4%) 1 (4%)
**17q** normal partial gain gain unknown	1 (4%) 10 (40%) 11 (44%) 3 (12%)
**ALK mutation** Yes no gain unknown	5 (20%) 16 (64%) 1 (4%) 3 (12%)
**CR or PR after induction chemotherapy** (3× N5/N6)	14 (56%)
HD + ASCT	20 (80%)
**Conditioning Regimen**	
Busulfan/melphalan	14/20 (70%)
Carboplatin/etoposide/melphalan	6/20 (30%)
**CD34+ cell dose** ×**10^6^/kg**, (range)	2.47 (0.59–21.73)
**Immunotherapy**	16 (64%)
**Time to immunotherapy**, d, (range)	137 (108–193)
**Event: progression or relapse**	7 (31%)
**Event: Refractory Disease**	3 (14%)
**Event: Toxicity**	1 (5%)
**Alive at last FU**	14 (56%)
**Median EFS**, year (range)	1.65 (0.11–3.67)
**Median follow-up OS**, year, (range)	2.14 (0.65–3.67)

Abbreviations:CR, complete response; PR, partial response; HD, high-dose; ASCT, autologous stem cell transplantation; FU, follow-up; EFS, event-free survival; OS, overall survival.

## Data Availability

The data presented in this study are available upon request from the corresponding author. The data are not publicly available due to privacy restrictions.

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
