# Peer review of "Immune Monitoring during Therapy Reveals Activitory and Regulatory Immune Responses in High-Risk Neuroblastoma"

_cancers, 2021, doi:10.3390/cancers13092096_

Round 1

Reviewer 1 Report

The authors performed immune monitoring of lymphocyte subsets and cytokines during treatment of a group of 25 children with high-risk neuroblastoma. The ideas and findings are novel and the paper is well written. A few minor revisions are suggested: 

There are studies showing that using GM-CSF alone with anti-GD2 may be as effective as using GM-CSF alternating with IL-2. The SIOPEN study also showed that using IL-2 (without GM-CSF) with anti-GD2 is equally effective with using anti-GD2 alone. Based on the findings in this study, would the author support using GM-CSF alone, IL-2 alone, or combined? 
Could the author propose a more specific potential treatment design and/or clinically feasible immune stimulating agents that can further enhance the immune function during maintenance therapy? 

Prolonged T cell dysfunction was observed in all patients. May the authors propose some specific strategies that can potentially improve the T cell function and treatment effect? 

Anti-GD2 has been used with re-induction chemotherapy (e.g. Irinotecan and Temozolomide) in relapsed/refractory patients. However, the NK and T cells may be much impaired during chemotherapy. Could the authors also discuss about this issue and propose some strategies that might enhance the immune function during re-induction chemotherapy in patents at relapse? 

Table 1. I would suggest to break each value of the variables into different lines. 
e.g. 
MYCN Neg. ___
Gain ___
Amp ___

Reviewer 2 Report

In Szanto, Cornel et al. the authors characterized the immune cells of neuroblastoma patients at diagnosis and during treatment. They showed that patient present with very heterogeneous fighting-capacity of the immune system which also reacts differently during therapy. The different reactions might explain differences in therapy outcome. Since the analysis is based on immune cells from peripheral blood it is easy to access for analyzing and implementing into the treatment plan. For me int is missing if their findings are related to outcome as well. p.e. do patient with a better immune reaction show improved therapeutic efficacy?

Overall I really like the study and I think the authors did a great job collecting all the patient data. 

Nevertheless I think showing each patient in a different color p.e. in Figure 1 doesn't add a lot of information but is rather confusing. In my opinion the authors should work on the presentation of the data to make it more intuitive. Also labeling should be enlarged to make it easier for the reader. All appreciations have to be mention in the figure legend (p.e. Dx, HD, IT).

Also please recheck the axes of your graphs. Since they are anyway not uniform I wonder, why p.e. in panel 1 E it goes till 1 if 0.5 is sufficient. This would also help showing the data more clearly. 

In most Figures (p.e. Figure 2) labeling is capitalized but in the figure legends it is mentioned in lowercase letters. This should be uniform throughout the paper. 

From Figure 2 on authors always compare the two states "pre" and "post". Why did the authors also include patients in this figure where only the "pre" value is available. Does this really add a lot of information? For me you can see that all patients present with a heterogeneous immune profile and most important message is the change upon treatment. 

Please check throughout the paper that special characters are presented correctly. (p.e. legend of Figure 4 says IFN-y).

Figure 6: I cannot grasp the statement the authors want to make with this Figure. What is the Reference Value marked in panel A. What is the function of the red arrows. 

Labeling in Figure 7 is too small. Delete all information you don't need (p.e. legend in panel E). In Panel D the authors included two gates but they did not explain what/how this gates where chosen. I think for the message they want to make they could simply delete the gates. 

Legend of Supplementary Figure 1 does not fit to what they show in the figure. According to the graph Patient 17 received ASCT. Please double check.
